# Genetic Alterations in Adult T-Cell Leukemia/Lymphoma: Novel Discoveries with Clinical and Biological Significance

**DOI:** 10.3390/cancers14102394

**Published:** 2022-05-12

**Authors:** Shugo Sakihama, Kennosuke Karube

**Affiliations:** 1Department of Pathology and Cell Biology, Graduate School of Medicine, University of the Ryukyus, Nishihara 903-0215, Japan; 2Department of Pathology and Laboratory Medicine, Graduate School of Medicine, Nagoya University, Nagoya 466-8550, Japan

**Keywords:** adult T-cell leukemia/lymphoma, human T-cell leukemia virus type I, genetic alterations, HTLV-1 *tax* subgroup

## Abstract

**Simple Summary:**

In recent years, many researchers have devoted their efforts to clarifying the genomic aberrations of adult T-cell leukemia/lymphoma (ATLL), and the overview of somatic alterations has become more apparent. Some study groups have also investigated the clinical significance of genetic alterations in ATLL to improve the therapeutic strategy, and have found key alterations related to pathogenesis. In addition, studies employing advanced technology have revealed the evolutionary processes of tumor cells and novel genetic alterations in ATLL. This review aims to provide readers with current clinical and biological knowledge regarding aspects of genetic alterations to help promote research in ATLL.

**Abstract:**

Adult T-cell leukemia/lymphoma (ATLL) is a refractory T-cell neoplasm that develops in human T-cell leukemia virus type-I (HTLV-1) carriers. Large-scale comprehensive genomic analyses have uncovered the landscape of genomic alterations of ATLL and have identified several altered genes related to prognosis. The genetic alterations in ATLL are extremely enriched in the T-cell receptor/nuclear factor-κB pathway, suggesting a pivotal role of deregulation in this pathway in the transformation of HTLV-1-infected cells. Recent studies have revealed the process of transformation of HTLV-1-infected cells by analyzing longitudinal samples from HTLV-1 carriers and patients with overt ATLL, an endeavor that might enable earlier ATLL diagnosis. The latest whole-genome sequencing study discovered 11 novel alterations, including *CIC* long isoform, which had been overlooked in previous studies employing exome sequencing. Our study group performed the targeted sequencing of ATLL in Okinawa, the southernmost island in Japan and an endemic area of HTLV-1, where the comprehensive genetic alterations had never been analyzed. We found associations of genetic alterations with HTLV-1 strains phylogenetically classified based on the *tax* gene, an etiological virus factor in ATLL. This review summarizes the genetic alterations in ATLL, with a focus on their clinical significance, geographical heterogeneity, and association with HTLV-1 strains.

## 1. Introduction

Adult T-cell leukemia/lymphoma (ATLL) is a peripheral T-cell malignancy caused by human T-cell leukemia virus type-I (HTLV-1) [1,2,3]. HTLV-1 infects individuals mainly via breastfeeding [4] and sexual intercourse [5], and then integrates its provirus into the host genome [6]. Although most HTLV-1 carriers remain asymptomatic, some develop ATLL and/or chronic inflammatory disease, such as HTLV-1-associated myelopathy (HAM) [7]. ATLL occurs in approximately 60 of 100,000 carriers per year, with a peak occurrence at approximately 70 years of age; the lifetime risk of ATLL onset has been estimated at 3–5% [8,9]. HTLV-1 encodes Tax on the plus-strand [10] and HTLV-1 bZIP factor (HBZ) on the minus-strand [11] of the *pX* region. These viral molecules contribute to persistent HTLV-1 infection in carriers and the development of ATLL [12,13].

Recently, several study groups have actively investigated somatic alterations in ATLL, leading to the clarification of the molecular mechanisms of tumorigenesis [14,15,16,17,18]. Some research has also elucidated the impact of alterations on clinical characteristics and prognosis [16,17,18,19]. In this review, we summarize the genetic alterations in ATLL and their associations with oncogenesis, clinical features, prognosis, and viral strains.

## 2. Diagnosis and Clinical Subtypes

ATLL is diagnosed with the combination of anti-HTLV-1 antibody positivity and evidence of T-cell malignancy based on cytopathological findings, including the cell surface phenotype and immunohistochemistry. A definitive diagnosis requires confirmation of the monoclonal integration of HTLV-1 provirus into abnormal lymphocytes by Southern blot hybridization [6]. We recently established a new diagnostic algorithm applicable to formalin-fixed paraffin-embedded tissue samples [20], which can prove viral infection in malignant cells, even when Southern blot hybridization cannot be performed. The immunophenotype of typical ATLL tumor cells is positive for CD3, CD4, CD5, CD25, CCR4, and CADM1, and negative for CD7 and CD8 [21,22,23,24]. In addition to these cell surface markers, ATLL cells often express Foxp3 [25], a master regulator for the differentiation of regulatory T (Treg) cells [26]. Patients with ATLL manifest various clinical symptoms, such as the infiltration of tumor cells into multiple organs in peripheral blood with the appearance of abnormal lymphocytes (flower-cells), lymph nodes, and skin, as well as opportunistic infections due to the immunodeficient status [27]. Shimoyama et al. [27] classified ATLL into four clinical subtypes based on clinical features and prognostic factors: acute, lymphoma, chronic, and smoldering. Patients with an aggressive form of the disease, namely acute, lymphoma, and chronic with unfavorable prognostic factors (abnormal levels of blood urea nitrogen, lactate dehydrogenase, or albumin), have a poor prognosis with a median survival time (MST) of approximately 1 year [28], even when treated with combination chemotherapy. The chronic type without poor prognostic factors and the smoldering type, which are called indolent ATLL, are usually followed up without chemotherapy until disease progression in Japan [29].

## 3. Treatment and Prognosis

In the largest retrospective analysis of 1594 patients with ATLL in Japan [29], the most common treatments for aggressive ATLL were CHOP (cyclophosphamide, doxorubicin, vincristine, and prednisolone), CHOP-like, or VCAP–AMP–VECP regimens (mLSG15; cyclophosphamide, doxorubicin, vincristine, ranimustine, vindesine, carboplatin, etoposide, and prednisone). In a phase-III randomized trial for aggressive ATLL, the mLSG15 arm showed a higher complete remission (CR) rate and better overall survival (OS) compared with the biweekly CHOP arm (40% vs. 25% for CR and 24% vs. 13% for OS at 3 years) [28]. Therefore, the mLSG15 regimen is the current standard chemotherapy for aggressive ATLL; however, a portion of patients are likely to be treated with the CHOP regimen because mLSG15 is more toxic and not applicable for elderly patients, which represent a major part of patients with ATLL [29]. Allogeneic hematopoietic stem cell transplantation (HSCT) is also performed as a curative treatment for aggressive ATLL [30]; it was performed in approximately 17% of patients in the nationwide survey [29].

CCR4, a member of the G protein-coupled receptor family [31], is highly expressed in ATLL cells [22]. The monoclonal antibody mogamulizumab targets CCR4 and eliminates tumor cells by antibody-dependent cellular cytotoxicity [32,33]. Mogamulizumab is applied for the treatment of relapsed/refractory ATLL in Japan based on the results from a multicenter phase II study [34]. Another phase-II trial exhibited higher CR rates in the mLSG15-plus-mogamulizumab regimen compared with chemotherapy alone as an initial therapy (52% vs. 33%) [35], while an increased risk of graft-versus-host disease-related mortality was reported in patients treated with mogamulizumab prior to allogeneic HSCT [36].

Antiviral therapy, a combination of zidovudine (AZT) and interferon-alfa (IFN) [37,38], is usually conducted worldwide, except in Japan [39]. A retrospective analysis evaluating the efficacy of first-line treatment on 254 patients with ATLL [40] showed superior 5-year OS on patients who received antiviral therapy than those treated with chemotherapy alone (46% vs. 20%, *p* = 0.004). Regarding clinical subtypes, antiviral therapy gave significantly longer OS to patients with leukemic subtypes (acute, chronic, and smoldering) compared to those with chemotherapy, whereas chemotherapy was more effective than antiviral therapy for patients with the lymphoma type. Of note, antiviral therapy for 17 patients with chronic or smoldering types showed 100% 5-year OS. Datta et al. [41] demonstrated that continuous AZT treatment reactivates p53 transcriptional activity, leading to the senescence of HTLV-1-infected cell lines, and patients carrying wild-type p53 responded to AZT treatment in their cohort. A reinstated p53 response triggered by a new histone deacetylase (HDAC) inhibitor was also reported in T-cell prolymphocytic leukemia cells [42]. Although p53 recovery can explain a pharmacological mechanism of AZT treatment in patients harboring wild-type *TP*53, the reason for survival disadvantage in the lymphoma type remains unclear. In Japan, a randomized trial of AZT and IFN versus watchful waiting for patients with favorable chronic or smoldering types (JCOG1111C) is currently in progress.

Sasaki et al. [43] reported the overexpression of the enhancer of zeste homolog 2 (EZH2), a component of polycomb responsive complex (PRC) 2 catalyzing the trimethylation of lysine 27 on histone H3 (H3K27me3), in ATLL cells. Western blotting showed clear H3K27me3 bands in most primary ATLL cell samples (10/13 cases), while faint bands were detected in peripheral blood mononuclear cells (PBMCs) from healthy donors (*n* = 6). Furthermore, the treatment of a histone methylation inhibitor (3-deazaneplanocin A) and a HDAC inhibitor (panobinostat) decreased the viabilities of ATLL cell lines, suggesting a potential therapeutic target of histone methylation in ATLL. The study combining microarray analysis and genome-wide chromatin immunoprecipitation assay [44] exhibited H3K27me3-associated downregulations in 54.6% of genes in ATLL cells. These genes included tumor suppressors (*NDRG*2, *CDKN*1*A*, *ZEB*1, and *BCL*2*L*11) and *CD7*, whose expression decreases with tumor cell progression. Recently, Yamagishi et al. [45] found that the EZH1/2 dual inhibitor (valemetostat) suppresses the proliferation of various cancer cell lines, including ATLL cells, more effectively than EZH2 inhibition alone. Repeated dosing of this agent did not show severe toxicity in rats for 14 days [46]. A phase-II study (VALENTINE-PTCL01) is currently underway.

Despite the advances in therapeutic strategies for ATLL, the outcomes remain unfavorable, especially in elderly patients, for whom intense treatment is not applicable. The identification of new therapeutic targets and the development of therapeutic agents are desirable.

## 4. Genetic Alterations in ATLL: Somatic Mutations

Kataoka et al. [14] conducted a comprehensive genomic study of 426 patients with ATLL in Japan and identified 50 frequently altered genes enriched in the T-cell receptor (TCR)/nuclear factor (NF)-κB pathway (*CD28*, *FYN*, *VAV1*, *RHOA*, *PLCG1*, *PRKCB*, *CARD11*, and *IRF4*), the T-cell trafficking pathway (*CCR4*, *CCR7*, and *GPR183*), immune-related genes (*HLA-A*, *HLA-B*, *CD58*), and other T-cell-related genes (*TBL1XR1*, *GATA3*, *NOTCH1*, and *STAT3*). Several research groups, including ours, have subsequently reported the genetic alterations in ATLL [16,17], and the mutational profiles have become more evident. Figure 1 shows the mutated genes and the types of mutations we have detected [17]. Notably, many mutated genes overlap with the Tax interactome [14,47], which includes molecules that interact with the Tax protein, and the gain-of-function and loss-of-function mutations in these genes could replace the role of Tax [14]. Although Tax contributes to various oncogenic cellular events, including the immortalization of cells [48,49], inhibition of the DNA repair mechanism [50], and disruption of the cell cycle checkpoints [51,52], its expression is suppressed in most patients with ATLL [14,53] due to strong immunogenicity [54]. These findings support the hypothesis that HTLV-1-infected cells compete with host immunity and survive by depending on Tax during the asymptomatic carrier phase, while acquiring genetic alterations allows their Tax-independent proliferation and immune escape, leading to ATLL development.

Recently, whole-genome sequencing (WGS) of 150 patients with ATLL revealed 11 altered genes unrecognized in previous studies that performed whole-exome or targeted sequencing [18]. Among these, the most frequently altered gene was *CIC*, which was affected by loss-of-function mutations (31%) or intragenic deletions (3%) in exon 2 (specific to the long isoform). CIC forms a complex with the transcriptional repressor ATXN1 [55]. Alterations in *CIC* or *ATXN1* were found in 53% of cases in a mutually exclusive manner [18], suggesting their crucial role in ATLL tumorigenesis. The *CIC* long isoform knockout mouse exhibited an increase in CD4^+^CD25^+^CD127^−^Foxp3^+^ T-cells in the spleen [18], indicating Treg or T-cell activation. Future studies are warranted to evaluate these newly altered genes for their relevance to pathogenesis and prognosis in ATLL.

## 5. Genetic Alterations in ATLL: Copy Number Alterations/Structural Variations

The following loci where copy number alterations (CNAs) frequently occur in the genome of patients with ATLL, and the genes presumed to be responsible for ATLL, have been reported: gains of 1p36 (*SKI*), 1q, 2q33.2 (*CD28*), 3, 6p25.3 (*IRF4*), 7p, 7q, 8q, and 14q32 (*BCL11B*), and losses of 1p13 (*CD58*), 6p21 (*HLA-A*), 6p22 (*HLA-B*, *ATXN1*), 6q14 (*SYNCRIP*), 9p21 (*CDKN2A*), 13q32 (*GPR183*), 14q31 (*NRXN3*), and 17p13 (*TP53*) [14,17,19,56,57,58,59,60]. In addition, structural variations (SVs) include *CTLA4*-/*ICOS*-*CD28* fusion genes; intragenic deletions in *IKZF2*, *TP73*, and *CARD11* [14]; 3′-untranslated region (UTR) disruptions of *PD-L1* [61]; and newly identified focal deletions of *REL* resulting in 3′-truncations [18].

Kataoka et al. [14] reported that 7% (7/105) of patients harbored the *CTLA4*-/*ICOS*-*CD28* fusion genes, whereas Yoshida et al. [62] found a higher frequency of these fusions in patients younger than 50 years old, albeit in a small cohort (38%, 3/8 cases). Tandem duplications were frequently observed in the region of 2q33.2 (*CD28*, *CTLA4*, and *ICOS*), which were inferred to be responsible for the *CTLA4*-/*ICOS*-*CD28* fusions [14]. These fusion proteins enhanced B7/CD28 co-stimulation activity by fusing the extracellular domain of CTLA4 or ICOS with the intracellular domain of CD28 [14,62]. Experiments with Ba/F3 cells cocultured with Raji cells expressing CD80 and CD86 demonstrated that Ba/F3 cells, in which CTLA4-CD28 expression was induced, were conferred cytokine-independent growth by recruiting p85α linked to the phosphorylation of AKT, a serine-threonine kinase, and extracellular single-regulated kinase (ERK) 1/2 [62]. It is suggested that the deregulation of CD28 signaling by these fusion genes is involved in the development of ATLL, especially in younger patients.

In a previous report, 9p24.1 (*PD-L1*) amplifications were correlated with the expression levels of *PD-L1* in classic Hodgkin lymphoma (cHL) and mediastinal large B-cell lymphoma [63]. Amplifications of 9p24.1 (*PD-L1*) have been associated with poor progression-free survival (PFS) in cHL [64] and favorable event-free survival and PFS in diffuse large B-cell lymphoma (DLBCL) [65]. In ATLL, 9p24.1 (*PD-L1*) amplifications have been detected in approximately 10% of cases [17,19] and correlated with elevated *PD-L1* expression [61]. Focal deletions, tandem duplications, inversions, and translocations have also been reported for the *PD-L*1 locus in ATLL, detected in 26.5% (13/49) of cases [61]. These SVs caused *PD-L1* 3′-UTR disruptions, regardless of the types of SVs, which contributed to *PD-L1* overexpression, independent of its amplification. In the in vitro experiments, cell lines with *PD-L1* 3′-UTR disruptions induced by CRISPR-Cas9 exhibited markedly elevated *PD-L1* expression [61]. Mice transplanted with EG7-OVA cells harboring *PD-L1* 3′-UTR disruptions displayed almost no cytotoxic T-cell infiltration and regression of transplanted cells, contrasting with observations in EG7-OVA cells with intact *PD-L1* [61]. These results indicate that *PD-L1* 3′-UTR disruptions can lead to tumor immune escape. Furthermore, these alterations have been observed in B-cell lymphoma and many other solid tumors [61], and might be a biomarker for the application of PD-L1/PD-1 inhibitors. However, there are reports of rapid progression in ATLL after treatment with the PD-1 inhibitor nivolumab [66,67]. Wartewig et al. [68] provided crucial findings on this adverse event. The chimeric interleukin-2-inducible T-cell kinase (ITK)-spleen tyrosine kinase (SYK) constitutively activates PI3K/AKT and PKC/NF-κB signaling [69]. PD-1 attenuates such signaling by enhancing the activity of the tumor suppressor phosphatase PTEN [70,71]. They demonstrated that *ITK-SYK*^CD4-creERT2^ mice with homozygous/heterozygous *PDCD1* deficiency were affected by the severe invasion of ITK-SYK^+^PD-1^−^CD4^+^ T-cells into the solid organs, indicating that PD-1 is a haploinsufficient tumor suppressor. Treatment with immune checkpoint inhibitors for *ITK-SYK*^CD4-creERT2^ mice with wild-type *PDCD1* induces the lethal proliferation of ITK-SYK^+^CD4^+^ cells as well as results in mice with *PDCD1* deletions, while treatment with the PI3K inhibitor idelalisib can extend the survival of these mice. These results indicated that PD-1 dysfunction and PD-L1/PD-1 inhibition could accelerate the expansion of T-cell malignancy with activated TCR signaling, such as ATLL. On the other hand, Ishitsuka et al. [72] argued that no patient had a similar clinical course after the PD-1 blockade in their Japanese cohort. These findings indicate that the further elucidation of the molecular background of the adverse effect of the PD-1 inhibitor is required before application to daily practice.

## 6. Integrated Analysis of Genetic Abnormalities

Several genes frequently affected by concurrent CNAs and mutations have been reported [14,17,18]. Figure 2A shows the genes in which we frequently identified homozygous deletions or heterozygous deletions and mutations, namely biallelic alterations [17]. Biallelic alterations of *CDKN2A* and *TP53* have been identified frequently in ATLL and seem to be closely involved in its development and progression. Although *CDKN2A* mutations are rare in ATLL, it is one of the genes in which homozygous deletions occur frequently [17,19,57]. On the other hand, biallelic inactivation in *TP53* seems to be attained by coexisting somatic mutations and heterozygous deletions, rather than homozygous deletions [14,17,19]. Our research also confirmed the mutually exclusive nature of biallelic alterations of both genes (Figure 2B) [17], a finding consistent with previous reports [57,73].

The latest WGS study updated the frequencies of the alterations in 32 major driver genes by integrated analysis of coding and non-coding mutations, SVs, and CNAs [18]. The most frequently altered gene was *CARD11*, followed by *PLCG1*, *CCR4*, *CIC*, *PRKCB*, *TBL1XR1*, *ARID2*, *CDKN2A*, and *TP53*, all with high frequency (>30%). This analysis revealed novel abnormalities not detected by whole-exome sequencing, providing a new overview of genetic alterations. These data will accelerate the elucidation of pathogenesis and the development of a new treatment for ATLL.

## 7. Synergistic Role of Alterations in the TCR/NF-κB Pathway

Recently, Kogure et al. [18] reported novel focal deletions in *REL*, which generate a C-terminally truncated form from exon 7 of c-Rel, a member of the NF-κB family, at a frequency of 13% (19/150 cases). These deletions are presumed to cause gain-of-function because the deleted regions are involved in the inhibitory domain. Subsequent luciferase reporter assays showed that truncated c-Rel enhanced NF-κB activity when co-expressed with RelA, compared with wild-type c-Rel, in HEK293T cells. Moreover, *REL* knockout inhibited the proliferation of the DLBCL cell line harboring *REL* SV, suggesting a function as a driver for truncated c-Rel [18].

Another study also reported the synergistic effects of mutations in *PRKCB* and *CARD11*, central molecules in the TCR/NF-κB pathway [14]. The coexistence of the E626K CARD11 mutant and the D427N PKCβ mutant increased NF-κB transcriptional activity compared with the expression of each mutant alone in HEK293T cells [14]. These findings suggest a synergistic effect of the two altered genes on the development and progression of tumor cells. Interestingly, *CARD11* alterations were distributed in multiple locations [14,16,17,18], such as the coiled-coil domain and the PKC-responsive inhibitory domain, in which intragenic deletions have been frequently identified [14,18]. In contrast, single-nucleotide substitutions in *PRKCB* were concentrated in the kinase domain [14,16,17,18]. Further analyses for detailed functional significance are required to elucidate the associations of *PRKCB* mutations and a variety of *CARD11* alterations.

## 8. Studies Using Longitudinal Samples

Several studies have revealed the evolutionary process of tumor cells in ATLL by comparing temporally different samples. Rowan et al. [74] performed a longitudinal study analyzing somatic mutations in blood samples from carriers to patients with overt ATLL. With deep sequencing, they detected mutations in the PBMCs of HTLV-1 carriers up to 10 years before ATLL development. There were numerous shared mutations between samples in the carrier status and the subsequent leukemic samples, and some premalignant clones already showed tumor-dominant HTLV-1 integration sites. These tumor-dominant HTLV-1-infected clones acquired additional mutations over time in carriers, and variant allele frequencies of mutations increased 6 months prior to diagnosis. Yamagishi et al. [75] developed a highly precise method for detecting mutations and HTLV-1-infected clones using single-cell analysis. They analyzed serial PBMC samples from asymptomatic carriers, patients with indolent ATLL, and those with disease progression. Their analysis showed the evolutionary process of ATLL tumor cells in terms of viral clonality, mutational composition, and transcriptome variation. The HTLV-1-infected cells maintained polyclonal populations without dominant clones in most carriers who did not progress to ATLL during follow-up periods. On the other hand, the carriers who developed ATLL harbored dominant clones expanded in a monoclonal or oligoclonal manner, and the clones already had several mutations (e.g., *PLCG1*, *VAV1*, and *STAT3*), which are findings consistent with those of a previous study [74]. These clones gained additional mutated genes (ex., *PRKCB*, *IRF4*, *TBL1XR1*, and *NOTCH*) at the time of ATLL onset or disease progression. Moreover, the transcriptome of malignant clones with *PRKCB* mutations in a patient with disease progression exhibited the upregulation of genes associated with cell growth (E2F and MYC targets). These studies [74,75] have provided direct evidence of the multistage oncogenesis of ATLL.

Marçais et al. [16] performed deep sequencing for PBMCs or tumor tissues from 61 patients with ATLL, including 15 serial cases who relapsed after remission or progressed from an indolent form. In 5/7 relapsed cases, malignant cells with the identical virus integration site at the time of diagnosis acquired de novo mutated genes involved in immune escape and the NF-κB/NFAT pathway (*PLCG1*, *PRKCB*, and *CARD11*). Viral clone switches and different mutational constitutions were observed in the other two cases, indicating the complexity of clonal dynamics over the course of treatment.

## 9. Heterogeneity of Genetic Alterations in ATLL

Figure 3 shows the mutational landscape of ATLL described in recent studies based on next-generation sequencing [14,15,16,17]. The somatic mutational profiles are similar in most genes among patients from mainland Japan [14], those with African and Caribbean origins [16], and those from Okinawa Prefecture in Japan [17]. Consistently in these profiles, mutations have been remarkably accumulated in the TCR/NF-κB pathway-related molecules (70–90% of cases) [14,16,17,18]. The T-cell trafficking pathway (*CCR4* and *CCR7*) has also been affected by gain-of-function mutations [14,76], generating truncations in the C-terminus at a frequency of 30–50% [14,16,17,18]. On the other hand, a research group in North America reported a higher mutation frequency in epigenetic and histone-modifying genes and a lower mutation frequency in genes of the TCR/NF-κB and JAK/STAT pathways, in contrast with other studies [15]. Moreover, the frequencies of mutations of several genes have also varied among studies (Figure 3) [14,16,17]. This regional heterogeneity in several genetic alterations in ATLL might be due to the characteristics of the patients. For example, the frequency of some mutated genes differed between clinical status (aggressive or indolent) and age (<70 or ≥70 years) [19]. The *PRKCB* and *TP53* mutation rates in our study [17] were different from those of other studies [14,16]. These mutated genes were more frequently identified in the aggressive rather than the indolent form [19], consistent with the fact that all patients in our study had the aggressive form. Marçais et al. [16] mentioned that there was no significant difference in mutated genes between African and Caribbean origins; however, there were more mutations in the cell cycle pathway, particularly *TP53*, in patients from Guyana (6/7 cases) compared with the other regions (4/26 cases for African origins and 6/22 cases for Caribbean origins). We previously reported that the frequencies of some altered genes were associated with HTLV-1 *tax* subgroups (HTLV-1-*tax*A and HTLV-1-*tax*B) [17], a phenomenon contributing to the geographical heterogeneity [77]. Germline mutations of *HAVCR2,* leading to the loss of function of TIM3, were proven to promote the disease development of subcutaneous panniculitis-like T-cell lymphoma [78]. Although no germline variants have been reported to be associated with some clinicopathological characteristics or the diseases progression, these factors would be objectives for future studies in ATLL. The differences in mutational profiles should be assessed accurately in consideration of these factors.

Regarding other T-cell lymphomas, a high frequency (50–70% of cases) of *RHOA* G17V mutations in angioimmunoblastic T-cell lymphoma [79,80] and the enrichment of mutations in JAK/STAT signaling in T-cell prolymphocytic leukemia (62.1% of 275 cases) [81] have been reported. Watatani et al. [82] revealed a landscape of genetic alterations in peripheral T-cell lymphoma not otherwise specified (PTCL-NOS) by comprehensive genetic analysis for 124 cases. They classified patients with PTCL-NOS into three molecular subtypes based on genetic alterations and immunophenotypes: (1) those characterized by alterations in T-follicular helper (TFH)-related genes (*TET2*, *RHOA*, and *IDH2*), (2) those harboring *TP53*/*CDKN2A* alterations, and (3) those other than the subtypes above. These molecular classifications will be useful in developing new therapeutic strategies and understanding the pathogenesis of tumors. Although ATLL has been clinically categorized by Shimoyama classification [27], the molecular features are highly heterogeneous. The establishment of molecular classification in ATLL is also required.

## 10. Associations of Genetic Alterations with Clinical Features

Kataoka et al. [19] investigated the impact of genetic alterations on the clinical characteristics and outcomes for 226 patients with ATLL. They found higher frequencies of *TP53* mutations, 13q32 (*GPR183*) losses, and 16q23 (*WWOX*) losses in patients aged >70 years old. Mutations in *PRKCB*, *TP53*, and *IRF4*, and deleted genes, including *CDKN2A* and *TP53*, were more common in patients with the aggressive form than in those with the indolent form, whereas *STAT3* mutations were enriched in the indolent form, suggesting the association of these altered genes with the clinical status. They also identified that *PRKCB* mutations and 9p24 (*PD-L1*) amplifications are poor independent prognostic factors in aggressive ATLL from known clinical prognostic factors: the Japan Clinical Oncology Group Prognostic Index (PI) [83] and age (<70 or ≥70 years) [84]. Regarding patients with the indolent form, the number of genetic alterations correlated with worse OS, and *IRF4* mutations, 9p24 (*PD-L1*) amplifications, and 9p21 (*CDKN2A*) deletions contributed independently to reduce OS. Their study group classified patients into two molecular groups, 1 and 2, based on genetic alterations detected by WGS [18]. Group 1 was characterized by alterations in molecules associated with upstream signal transduction in the TCR/NF-κB pathway (*PLCG1*, *VAV1*, *CD28*, and *RHOA*), *STAT3* mutations, and fewer alterations than group 2. Group 2 comprised altered genes clustered in the downstream molecules of the TCR/NF-κB pathway (*PRKCB* and *IRF4*), immune-related molecules (*HLA-A*, *HLA-B*, and *CD58*), and epigenetic regulators (*EP300* and *TET2*). Interestingly, almost all cases with the lymphoma type belonged to group 2, the chronic type tended to be classified into group 1, and the frequencies of the acute and smoldering types were similar in both groups. Patients classified into group 2 showed significantly higher serum calcium, soluble interleukin-2 receptor (sIL-2R), and lactate dehydrogenase levels; lower albumin level; and shorter OS than patients in group 1, although there was no significant difference between survival estimates in each clinical subtype.

Yoshida et al. [57] analyzed serial samples from patients with the chronic and acute types, as well as those with disease progression from the chronic type. They reported that homozygous losses of 9p21 (*CDKN2A*) are characteristic of the acute rather than the chronic type and likely lead to disease progression. They also identified that losses of *CD58* and genes involved in the cell cycle, including *CDKN2A* and *TP53*, are associated with disease progression. Marçais et al. [16] showed worse PFS and OS in indolent ATLL patients harboring mutations of TCR/NF-κB-related molecules, indicating their contribution to aggressive progression.

Sakamoto et al. [85] described the clinical value of identifying *CCR4* mutations as an indicator of the good therapeutic sensitivity of mogamulizumab, although validation studies are warranted to confirm the practical use of testing for *CCR4* mutations as a solid biomarker. They also reported the clinical significance of alterations in *TP53* and *CD28*. Patients with *TP53* alterations had a significantly poorer Eastern Cooperative Oncology Group performance status, higher sIL-2R and serum calcium levels, and worse OS than those without alterations [86]. Other clinicopathological studies have suggested associations of *CD28* alterations with its overexpression, infiltration of CD80/CD86-positive non-malignant cells into the microenvironment, poor prognosis [87], and younger age of patients [88].

Our study group investigated genetic alterations in 51 genes significantly affected in ATLL and other lymphomas by using targeted sequencing and a single nucleotide polymorphism (SNP) array on 89 patients with aggressive ATLL [17]. We reconfirmed the negative impact of *PRKCB* mutations on OS (Figure 4A), a finding consistent with a previous study [19]. This reproducibility between different cohorts is a significant finding. Furthermore, the combined analysis of mutation and CNA data suggested the relevance of biallelic alterations in *PRDM1* to reduce OS in patients (Figure 4B). *PRDM1* biallelic inactivation was reported to be a poor prognostic factor in DLBCL [89], indicating the importance of integrated analysis of genetic alterations. These altered genes are associated independently with poor OS, confirmed by multivariate analysis, even with ATLL-PI [84], which is known to be a solid prognostic factor.

Cutaneous lesions are one of the characteristic manifestations of ATLL [27]. Cutaneous involvement is categorized into patch, plaque, multipapular, nodulotumoral, erythrodermic, and purpuric [90,91]. These types of lesions are reported to be associated with the clinical course [29,90,91]. Several researchers have proposed the cutaneous type as a new entity, in addition to the classical Shimoyama classification [27], based on their clinical and biological features [92,93]. Notably, Miyata et al. [93] revealed that there were distinct genomic profiles between erythema/papule and nodule/tumor types in skin lesions of patients with ATLL. However, details regarding the genetic alterations of cutaneous involvement and their clinical significance remain unknown. The heterogeneities of genetic alterations in tumor cells across organs should be elucidated.

Although many researchers have made an effort to elucidate ATLL pathogenesis, the roles of genetic alterations in clinical states are not fully understood. Given the variations in the characteristics of patients and the kinds of specimens that have been collected among the cohorts, more evidence needs to be accumulated. Of note, analysis that takes into account disease subtype and comparison with known prognostic factors is recommended [83,84].

## 11. Phylogenetic Classifications of HTLV-1 Strains

There are an estimated 10–20 million HTLV-1-infected individuals in the world, and the main endemic areas are Central Africa, South America, the Caribbean coast, Melanesia, and Southwestern Japan [77]. HTLV-1 is classified into four major subtypes, namely Cosmopolitan subtype A, Central African subtype B, Central African/Pygmies subtype D, and Australo-Melanesian subtype C, based on phylogenetic analysis of the 5′-long terminal repeat (LTR) sequence [77,94,95,96,97]. Cosmopolitan subtype A, which is widely distributed worldwide, is further subdivided into four subgroups: Transcontinental, Japanese, West African, and North African [77,98]. The nucleotide sequence of the *tax* gene has also been employed in phylogenetic classification. Furukawa et al. [99] identified that HTLV-1 in Japan comprises the *tax*A subgroup (HTLV-1-*tax*A) and the *tax*B subgroup (HTLV-1-*tax*B), corresponding to the Transcontinental and the Japanese subgroups, respectively.

We previously surveyed the *tax* subgroup in 29 HTLV-1 carriers, 74 patients with ATLL, and 33 patients with HAM in Okinawa Prefecture [100]. We found 60 cases with HTLV-1-*tax*A (44%) and 76 cases with HTLV-1-*tax*B (56%), which was different from the distribution of those in mainland Japan (11% for HTLV-1-*tax*A and 89% for HTLV-1-*tax*B) reported by Furukawa et al. (Figure 5) [99]. Residents of Okinawa Prefecture have a historically and genetically distinct background from those of mainland Japan [101,102], a factor that might underlie this difference.

## 12. Associations of HTLV-1 Strains with Genetic Alterations

There are four specific nucleotide substitutions (positions 7897, 7959, 8208, and 8344) between HTLV-1-*tax*A and HTLV-1-*tax*B [99]. These nucleotide changes are reflected in amino acid substitutions in the leucine zipper region and the ATF/CREB-activating domains of the Tax protein [103] and the bZIP domain of HBZ [104]. It was reported that HTLV-1-*tax*A carriers had a 2.46-fold higher odds ratio for developing HAM than HTLV-1-*tax*B carriers, regardless of the HLA haplotype [99]. In PBMCs from patients with HAM, the expression levels of HBZ were higher in cases with HTLV-1-*tax*B than HTLV-1-*tax*A, whereas the expression levels of Foxp3 were higher in cases with HTLV-1-*tax*A than HTLV-1-*tax*B [104]. Foxp3 expression has been associated with the immunodeficient state in ATLL [25]. These observations suggest that variations in the *tax* subgroups affect the expression of viral genes and host genes and contribute to the development of HAM and ATLL. However, the effect of differences in the HTLV-1 strains on tumor cell biology has not been assessed sufficiently. Given the accumulation of genetic alterations in the Tax interactome of patients with ATLL [14,47], we hypothesize that the *tax* subgroups are associated with the genetic alteration profile of ATLL. In our recent targeted sequencing study [17], mutations in *GATA3*, a master regulator of Th2 cell differentiation, and *RHOA*, a member of the small GTPase Rho family, were significantly and more frequently detected in patients with HTLV-1-*tax*A compared with those with HTLV-1-*tax*B (Figure 6). A comparison of the genetic alteration profiles between patients with ATLL in Okinawa Prefecture and those in mainland Japan [14] showed that both mutated genes tended to be more frequent in cases from Okinawa Prefecture (Figure 3). Interestingly, in vitro experiments confirmed that Tax suppresses *GATA3* expression via ZEB, a repressor protein of the *GATA3* promoter [105], consistent with its mutation context in patients with ATLL in which truncating mutations, leading to loss-of-function, represent the majority [14,16,17,18]. Co-immunoprecipitation assays demonstrated the direct binding of Tax with RHOA, although the functional connection remains unclear [106]. Based on these findings, we assumed that HTLV-1 strains are associated with some altered genes, but before solid conclusions can be drawn, it is necessary to eliminate the possibility that the variety of clinical/genetic backgrounds of patients and the subtle differences in analytical methods among studies are the source of these alterations. Furthermore, functional analysis of Tax and these mutated genes and studies to obtain reproducibility in other HTLV-1 endemic areas is warranted, especially in regions where several HTLV-1 strains are concurrently present, such as Okinawa Prefecture.

## 13. Conclusions

We have reviewed the latest available information regarding genetic alterations in ATLL, focusing on discoveries made by employing state-of-the-art technology, such as WGS and single-cell analysis, as well as the relevance of genetic alterations with clinical findings and HTLV-1 strains. We expect that elucidation of the molecular pathogenesis of ATLL will be accelerated by accumulating data from multidimensional analyses using time-series and multi-organ specimens and advanced technology, including NGS, single-cell analysis, and proteomics [107].

## Figures and Tables

**Figure 1 cancers-14-02394-f001:**
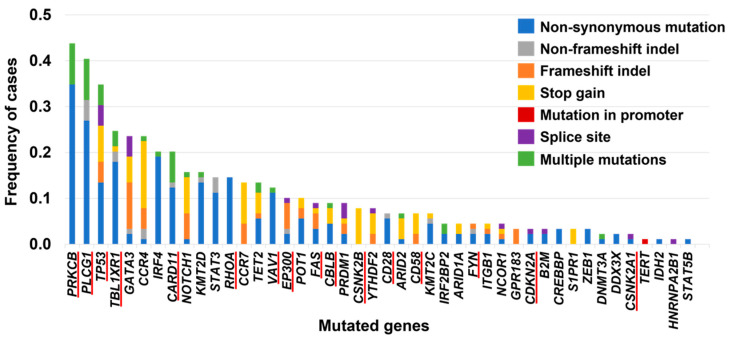
Mutated genes and their frequencies in aggressive ATLL identified in our previous study [17]. Genes underlined in red are part of the Tax interactome [14,47].

**Figure 2 cancers-14-02394-f002:**
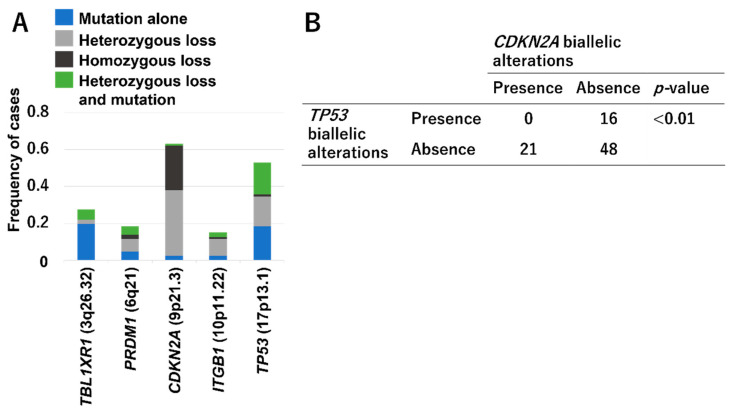
(**A**) Genes with significant biallelic alterations in ATLL based on our published data [17]. (**B**) Mutually exclusive nature of biallelic inactivation of *TP*53 and *CDKN*2*A*. The *p*-value was calculated with two-sided Fisher’s exact test.

**Figure 3 cancers-14-02394-f003:**
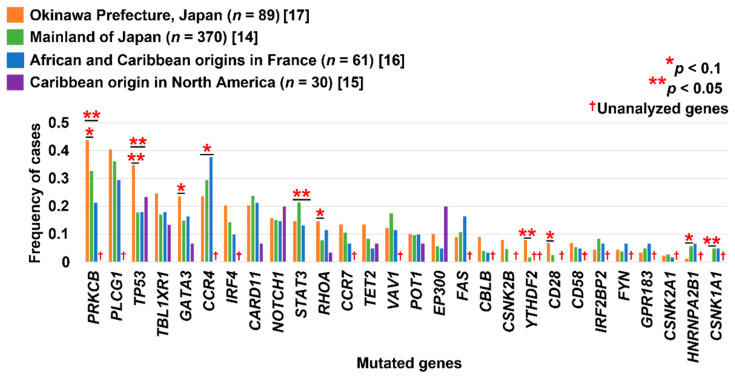
Heterogeneity of mutational profiles depending on the geographical region [14,15,16,17]. The *p*-value was calculated with two-sided Fisher’s exact test.

**Figure 4 cancers-14-02394-f004:**
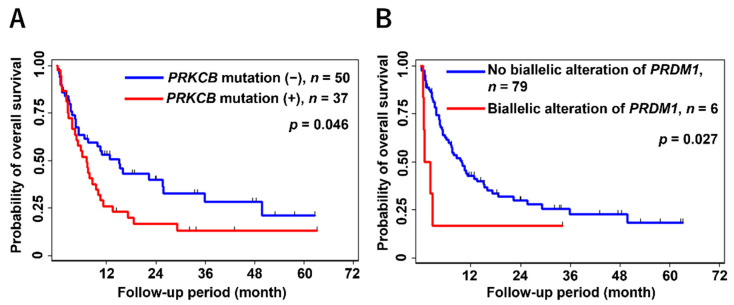
Kaplan–Meier survival curves for *PRKCB* mutation (**A**) and biallelic inactivation of *PRDM*1 (**B**) in aggressive ATLL [17]. Statistical significance was evaluated with the log-rank test.

**Figure 5 cancers-14-02394-f005:**
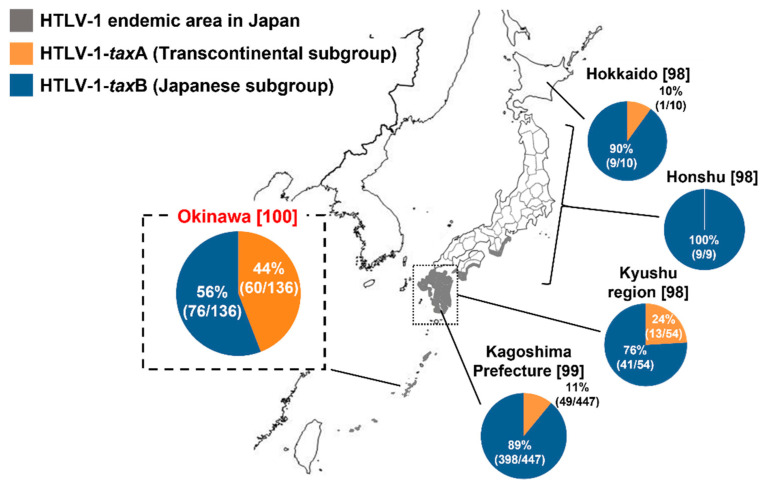
Distributions of HTLV-1 strains in Japan described in previous reports [98,99,100].

**Figure 6 cancers-14-02394-f006:**
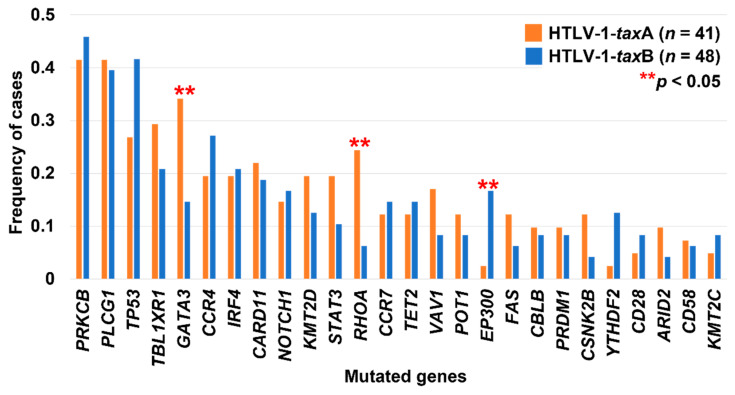
Associations between mutations and *tax* subgroups observed in our previous study [17]. *p*-values were calculated with two-sided Fisher’s exact test.

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
