# Peer review of "Genetic Alterations in Adult T-Cell Leukemia/Lymphoma: Novel Discoveries with Clinical and Biological Significance"

_cancers, 2022, doi:10.3390/cancers14102394_

Round 1

Reviewer 1 Report

In this review article, Sakihama and Karube comprehensively summarize the current knowledge of the genetic basis of adult T-cell leukemia/lymphoma (ATLL). Considering the significant increase in genetic data in ATLL, the article very nicely provides an overview of the clinical significance, geographical heterogeneity and association with HTLV-1 strains and disease progression. This summary is an important resource for both basic researchers and clinicians to understand the molecular basis of the disease and also guide further research efforts. Thus, this article is highly relevant to be published. The article is well written and covers the most relevant literature to the topic. Minor comments described below should be addressed before publication.

Minor comments:

  • Section 3, treatment and prognosis: anti-viral therapy (zidovudine and IFN-alpha) should be mentioned as an option for some patients with leukemic subtypes (PMID: 21672246).
  • Line 89: eligibility for allogeneic stem cell transplantation should not only be defined by age. I recommend inclusion of comorbidities. In fit patients, transplantation can be possible even beyond the age of 65 years.
  • Figure 1: it would be beneficial to include the findings of the whole-genome study of Kogure et al in this figure, as mutations in CIC are very frequent in their study.
  • Line 146: regarding the findings of Yoshida, please emphasize the very small cohort (8 pts.).
  • Line 175 / 176: Here it is important to add the findings of Wartewig et al. describing PD-1 as a haploinsufficient tumor suppressor (Nature 2017), which might explain hyperprogression upon PD-1 inhibition! Importantly, this is also supported by genetic findings in ATLL! Please add this to the discussion.
  • Line 279 – 288: I think this should be moved to section 10 (association of genetics and clinical features)

Reviewer 2 Report

 A section describing the different clinical subtypes of ATLL will be useful. Further, a section can describe the shared and distinct genetic mutations of these subtypes.

  1. A meta-analysis of somatic mutations  for TPLL detected across different studies has been published earlier. It could be a good resource to cite (; https://doi.org/10.3390/cancers11121833.)
  2. Is there any germline mutation or variant that affects the risk of ATLL? A description of these variants can help to understand the otherwise neglected contribution of germline variants on cancer risk and progression?
  3. A short paragraph about the role of identified somatic changes on drug response and resistance could extend the readership of the review. Also, a description of the evolving and upcoming treatments of the disease could be helpful. https://doi.org/10.1038/s41375-020-0772-6 could be a good citation here.

Reviewer 3 Report

As this work is a review and not an original study, I would like to do some comments. However they are quite fundamental. 

  1. First of all in the section 3 -Treatment and Prognosis, you have to extend available information. Mainly to add data on NCCN and ESMO guidelines, compared them with Japan guidelines and real-world practice. Data you provided on treatment is scarce.
  2. Methylation pathway genes, like TET2, IDH2 and DNMT3 are also altered in ATLL. Polycomb dependent repression is enhanced in ATLL by histone lysin 27 trimetylation and affects  almost half of ATLL genes. EZH2 and some other components of PRC2 complex are unregulated in ATLL. From therapeutic point of view EZH2 inhibitors could be potential treatment option. Also, there is nothing about some other tumor suppressor genes, like CDKN1A, BCL2L11 ecc. I believe that it is necessary to add and describe
  3. References - 80% of your references are quite old. There are a lot of new data on the field of proposed review. 

Round 2

Reviewer 2 Report

None

Reviewer 3 Report

After comments made during first revision and authors correction, this review could be accepted for publication